# Bio-Producing Bacterial Cellulose Filaments through Co-Designing with Biological Characteristics

**DOI:** 10.3390/ma16144893

**Published:** 2023-07-08

**Authors:** Roberta Morrow, Miriam Ribul, Heather Eastmond, Alexandra Lanot, Sharon Baurley

**Affiliations:** 1Materials Science Research Centre, Royal College of Art, London SW11 4NL, UK; miriam.ribul@rca.ac.uk (M.R.);; 2CNAP-Department of Biology, University of York, Wentworth Way, Heslington, York YO10 5DD, UK; heather.eastmond@york.ac.uk (H.E.); alexandra.lanot@york.ac.uk (A.L.)

**Keywords:** circularity, bacterial cellulose, co-design, bio-manufacturing, biomaterials, bio-design

## Abstract

The need for circular textiles has led to an interest in the production of biologically derived materials, generating new research into the bioproduction of textiles through design and interdisciplinary approaches. Bacterial cellulose has been produced directly from fermentation into sheets but not yet investigated in terms of producing filaments directly from fermentation. This leaves a wealth of material qualities unexplored. Further, by growing the material directly into filaments, production such as wet spinning are made redundant, thus reducing textile manufacturing steps. The aim of this study was to grow the bio-material, namely bacterial cellulose directly into a filament. This was achieved using a method of co-designing with the characteristics of biological materials. The method combines approaches of material-driven textile design and human-centred co-design to investigate co-designing with the characteristics of living materials for biological material production. The project is part of a wider exploration of bio-manufacturing textiles from waste. The practice-based approach brought together biological sciences and material design through a series of iterative experiments. This, in turn, resulted in designing with the inherent characteristics of bacterial cellulose, and by doing so filaments were designed to be fabricated directly from fermentation. In this investigation, creative exploration was encouraged within a biological laboratory space, showing how interdisciplinary collaboration can offer innovative alternative bioproduction routes for textile filament production.

## 1. Introduction

The textile industry is one of the main carbon polluters and generators of waste. In order to reduce carbon emissions and align with the 2050 UN/EU goals, the textile industry must change from linear to circular models [1]. For this reason, there is a demand for circular materials and manufacturing innovations. Within a circular model, the stages of material choice, manufacturing, use and disposal have direct influence on each other across social, environmental and economic sustainability [2]. Although there can be points of intervention at all these stages, this paper focuses on the raw material and production stages. 

The European Union has stated that waste should be regarded as a resource to transition material consumption from linear to circular modals; this is specifically with a view to reducing the reliance on non-renewables as raw materials [1,3]. In response to this, designers, scientists and textile engineers have looked to biology to utilise waste as a feedstock for bio-manufacturing materials. Bacteria, fungi and algae are at the forefront of bio-manufacturing investigation, either as separate materials or as multi-organism composites [4]. The abundance of natural characteristics, including self-formation, make these materials unique and desirable as potential self-forming textile structures [4,5]. The understanding of these characteristics has in turn generated theories of co-designing with the living materials to directly produce textiles [5,6]. Although co-design is an approach found in human-centred approaches, the field of leading and learning with biology has created new opportunities in which co-design can be applied to both biological and human design interactions. 

Bacterial cellulose is one of the main materials being investigated within the biomaterial field due to its unique characteristics such as its ability to self-assemble, its high mechanical strength and how it can be produced from accessible food sources or waste [7]. Most investigations of bacterial cellulose intended for apparel application produce the materials as a biological nonwoven sheet [7]. Research into producing BC into filaments is so far investigated through a process called wet spinning to create man-made cellulose fibres (MMCFs) [8]. Although this is an established and scalable way of producing filaments from cellulose, it requires the cellulose to first self-assemble then to be broken down to a dope and reshaped into a filament [8]. This adds a manufacturing step that may not be needed if the material can form itself, which further changes the inherent characteristics of the material. There is a gap in research regarding approaches of co-designing with bacterial cellulose characteristics in order to directly bio-manufacture bacterial cellulose with textile properties.

Our background review provides context in four parts: the relation between biomaterials and waste, how bioproduction innovations investigate co-design when translating biomaterials into textiles, a theoretical investigation of bacterial cellulose and how bacterial cellulose is explored in textile design. 

### 1.1. Background

#### 1.1.1. Circularity and Waste

In order to reduce the reliance on non-renewable resources, waste has been highlighted as an important circular feedstock for textiles and apparel [9,10,11]. Regenerating biomass waste into new materials is an area of research being investigated through biological interactions. Microorganisms such as bacteria and fungi can use waste as a form of feedstock to grow pliable materials investigated in areas such as architecture and material innovations [5,7,12]. Because these materials are biologically derived, they seldom use landmass in their production creating a circular raw material that has been derived from waste [11,12]. However, in order to manufacture these biomaterials into usable textile apparel, new creative manufacturing is required. Biologically derived cellulose has been proven to be produced from multiple different feedstocks including waste materials such as agricultural and food waste [13,14]. In contrast to other forms of cellulosic raw materials such as cotton, bacterial cellulose does not use landmass in its growth [13,15]. This makes bacterial cellulose an interesting material with circular potential. Although this paper investigates bacterial cellulose derived from glucose within the methodology, the methods are directly transferable to bio-waste-derived bacterial cellulose, such as crop residues and the organic fraction of municipal waste. It is the latter that is investigated by the co-authors of this article within the BBSRC Bio-manufacturing Textiles from Waste project.

#### 1.1.2. Bio-Producing Textiles

Several themes of bioproduction for textiles have been explored through the interdisciplinary interaction of design, science and textile manufacturing knowledge. Examples of innovative bio-manufacturing and design will be discussed, in particular the impact of co-creating with biomaterial characteristics.

Camere; Karana (2018), discuss theories of designing with materials and how the practices of working with materials can implement cleaner production systems [6]. Although the theoretical analysis of working with biomaterial is captured as a growing design method, the process works with the properties of the material, suggesting processes are co-designed. Without directly categorising works as co-design, the following examples sit within the intersection of design biology and technology.

Collet’s ‘bio lace’ (2012) is an example of leading with a conceptualisation of a design future, with synthetic biology emerging as a form of bio-production [16]. Situated in 2050, the themes of co-designing with living materials are captured, showing a future where bio products/waste (in this case the roots of edible plants) are grown directly into lace [16]. Bio lace captures the theory of designing with biology and provides a futuristic manufacturing concept that utilises waste as a resource. Translating this concept into physical garments is the work of Zena Holloway (2022) [17]. Holloway captures the themes discussed in bio-lace through physical interruption of biological systems. The material is not yet wearable but shows the themes of design and biology interlacing.

The trajectory of designing with living materials has expanded across multiple fields of interest. Alima, Snookand and McCormack (2022) work primarily with mycelium, furthering the concepts of co-creations and merging the boundaries of technology and biological material [18]. Their work discusses robotics and biology to create Bio-scaffolds, co-created through two seemingly opposing processes. Similarly, within architectural biomaterials, Scott et al. (2021) push the boundaries of digital knitted textile form and use it as a scaffold for biomaterial growth, designing with the self-formation of biomaterials [4,5]. This piece of research shows the inherent ability of knitted structure and scale, which arguably shows that the interactions of textile manufacturing has accelerated the scale of production of biomaterials. Rich in hybrid knowledge, the research of Alima, Snook and McCormack (2022), and Scott et al. (2021) is based in material architecture application. It shows how manufacturing of biological systems for material design can be intertwined with modern technologies. 

A hybrid process is also clear in Modern Synthesis’s interaction of biological matter, textile manufacturing and design knowledge [19]. The reinforced technical material is bio-manufactured from bacterial cellulose and a customisable woven technique, to create reinforced cellulose sheets. This material is distinct from other cellulose based textiles in being identified as a new class of nonwovens, further paving the way for new material definitions [19,20]. The design of the manufacturing has not only been directly influenced by the inherent characteristics of bacterial cellulose, but also structurally informed by textile engineering technologies [19,20]. The success in this case of the hybrid input of design, biological science and textile manufacturing suggests the importance of combined knowledge for the future of bio-manufacturing routes for textiles.

The commonality across these innovations is twofold. Firstly, the success of interdisciplinary collaborations and knowledge shows the necessity for designers and scientists to directly interact. Proving this theory, Ribul et al. (2021) discuss the material-driven textile design (MDTD [21]) methodology. This method demonstrates how the interaction between designers and scientists produces innovative circular material design outcomes informed by the properties of materials in scientific development. Although this method does not explore living material characteristics, which is adapted in the method section of this paper, the importance of recognising design within a lab space is captured. Secondly, there is a common theme of using scaffolds in the co-creation with biomaterials and textile technologies to provide a platform for the material to be manufactured in a sustainable way. The designers draw on the inherent qualities of the material to co-design a manufacturing process which directly embeds the biomaterial characteristics. Working with the characteristics of a biological material points to the requirement of a method to co-design with the biomaterial characteristics.

By utilising biomaterials’ characteristics of self-assembly, the energy of manufacturing steps is reduced, thus cutting out parts of the value chain [9]. The process of co-designing with the characteristics of a material provides an opportunity to cut down on conventional manufacturing steps for different material industries. Although bacterial cellulose designs have so far been focused on sheet forms embodying nonwovens structures, there is little or no investigation into how the concepts of co-design and bio-manufacturing can directly form filaments. This leaves a wealth of material applications unexplored.

#### 1.1.3. Bacterial Cellulose

Bacterial cellulose (BC) was the primary material under investigation in this study. The following section will facilitate an understanding of BC as a material.

Bacterial cellulose was first discovered by Brown in (1886), and by the 1990s, 100–150 tons were being produced per year [22,23]. Since then, bacterial cellulose production has increased due to the demand from industries: biomedical materials, packaging, cosmetic products and more recently textile applications [23,24,25]. Although the demand is increasing, bacterial cellulose can be grown from waste or food sources reducing the reliance on raw materials, further discussed in Section 1.1.4 [13,14]. 

Bacterial cellulose is the purest form of cellulose found within nature and is produced during fermentation by specific bacteria [26,27]. During the fermentation, which requires access to oxygen, sugar and nutrients, the bacteria secretes cellulose in the form of fibrils and ribbons [26,27]. These ribbons tangle together forming a pellicle at the interface of the liquid and oxygen [26]. It is this three-dimensional nanostructure that provides BC with its unique mechanical properties [25,26,27]. The entangled fibres form a mat-like material, mimicking a biologically formed nonwoven sheet of material. This unique physical strength changes depending on its wet or dry states as the structure changes through interaction with liquids [28]. Interestingly, air-dried pure BC has the texture of paper. 

There are different fermentation methods for growing bacterial cellulose such as static and shaken fermentations. For static fermentation, the liquid media is incubated in shallow trays, and the cellulose grows as an intertwined mat of cellulose microfibres at the surface [29]. Shaken fermentations are carried out in conical flasks incubated in a shaking incubator and produce spherical beads of various sizes depending on the conditions [27,29]. This is evidence of how changing the physicality of the fermentation can have a large impact on how the material grows. 

Chemically, bacterial cellulose is the same as plant-based cellulose; however, the crystallinity of the material is higher [30]. Bacterial cellulose has none of the impurities that plant-based cellulose has, such as hemicellulose or lignin and pectin. The microfibrils within bacterial cellulose have a cross sectional area of around 50 nm, giving BC a higher tensile strength than steel. These fibrils form long entangled matrices held together through interaction at the nanometre scale. This contrasts with cellulosic materials such as cotton textiles, where the material strength arises from interactions of whole cellulosic plant cell walls with radii of greater than 50 µm; hence, the material cannot be produced as a textile in the same way.

By understanding the biological characteristics of this growth, bio-production opportunities can be explored. If the growth of the material is a core factor in the production, such as growing the material directly to product, manufacturing steps could be reduced. The more steps in the manufacturing that are removed, the higher the sustainable prospects.

#### 1.1.4. Waste-Produced Bacterial Cellulose

Agricultural waste is a source of interest for alternative material production and has been reviewed as a viable feedstock for bacterial cellulose [13,14,31]. By using waste streams as a raw material input, the creation of new textiles focuses on utilising a material that is abundant. Enzymatic biology is used to convert cellulosic waste into sugars; this is then used within a fermentation process with bacteria [10,32]. This method of cellulose production has been proven at the lab scale and is being investigated as a recycling option for cellulosic waste streams [32]. BC has been produced by different forms of waste from sugarcane molasses, cashew tree residues and potato peel wastes, among others, utilising the nitrogen within the material [13,14,33,34]. Although these forms of waste may not be abundant sources, they show that BC can be produced from multiple feedstocks. There must be a shift from the use of non-renewable materials to utilising materials that are already within the system. This paper investigates bacterial cellulose derived from glucose within the methodology. This was chosen as the production itself was being investigated, and not the waste source. Nevertheless, the methods discussed within this paper are directly transferable to bio-waste-derived bacterial cellulose.

#### 1.1.5. Bio-producing Bacterial Cellulose for Textiles

This section discusses how bacterial cellulose is being investigated as a textile. Bacterial cellulose was suggested as a new material textile for the apparel industry by Suzanne Lee [11,35]. The material’s inherent characteristics are not suitable for apparel without adding further processing steps [36]. Recent investigations into BC as a material have focused on the material as a nonwoven self-assembling bio-structure. In particular, growing BC directly into shapes has been of interest in terms of designability. From an apparel perspective, this characteristic can be used to reduce waste and manufacturing steps such as garment cutting [37]. As discussed in the bio-manufacturing section above, self-assembly of BC has also been investigated by Modern Synthesis, who integrated weaving technologies with bacterial cellulose growth to create a new class of material [19,20]. These investigations are all centred on bacterial cellulose production as a sheet or nonwoven, leaving a gap, namely the investigation of how bacterial cellulose could be formed into other textile structures with alternative characteristics.

There is evidence that bacterial cellulose can be extruded into filaments for yarns through wet spinning [8,38]. By applying bacterial cellulose to filaments, the material is opened to a different type of textile production including knitted form. However, by wet spinning bacterial cellulose, the material is grown, then solubilised, then regenerated, adding potentially unnecessary manufacturing steps. This highlights a further gap requiring investigation: how co-designing with the biological characteristic of bacterial cellulose can produce filaments.

## 2. Method: Bioproduction of Bacterial Cellulose Filaments

This collaborative research formed part of the BBSRC Bio-Manufacturing Textiles from Waste project. The interdisciplinary design and biological sciences aspects of the research project are described in this methodology, which informed the investigation for co-designing with the inherent characteristics of bacterial cellulose. We first describe the method of co-designing with biology, informed by both the material-driven textile design (MDTD) methodology and human-centred co-design. We then describe the materials and methods for the production of the bacterial cellulose filaments.

### 2.1. Method for Co-Design with the Inherent Characteristics of BC

In order to inform the interdisciplinary investigation, the research involved a design placement in the laboratory at the University of the York over six months. This laboratory was set up solely for bacterial cellulose production. The first stage of the research consisted of the designer and lead author of this article to observe the static bacterial cellulose production methods and to learn about standard processes within a laboratory setting. As the experiments were developed iteratively, a broader understanding of the material’s characteristics was formed. From this, it emerged that in designing the production of filaments, the material’s characteristics are a key factor.

This paper combines approaches of two design methods to encourage designers to interact with the characteristics of biological materials. These methods have been selected as part of this research as they both offer approaches for working in interdisciplinary design–science collaboration and could be applied to biomaterial development. The first one is the material-driven textile design (MDTD) methodology [21]. The MDTD methodology encourages designers to have an active role within a laboratory for material experiments that begin with the material properties [21]. Ribul et al. (2021) demonstrate how design at the raw stage of materials can inform new circular material design processes that respond to the properties of materials and result in a reduction in processing steps [21]. The methodology is structured into three action steps in the development of new material design processes: exploration, translation and activation. The second method is co-design. Co-design is an approach discussed mainly within human interactions to encourage communication and collaboration for design strategies that best suit the needs of the user [39]. Combining approaches of co-design with MDTD would allow designers to co-design with the biological characteristics of living materials. This creates an opportunity to centre the biomaterial in design decisions for production.

Bacterial cellulose was first considered from a theoretical perspective that employed a literature review in order to gain understanding of its inherent characteristics and current production methods, which were then explored in the laboratory. By working directly with the material within the lab, we explored factors such as the fermentation process, the different requirements and conditions for growth, as well as opportunities for design intervention to edit the process. The interdisciplinary approach allowed for dynamic discussion around questions throughout the experimentation, as well as during planning and the review of results. Co-designing with the material highlighted three physical designable characteristics: first, the interaction of oxygen and nutrients allowing for self-assembly; second, the physical parameters of growth and vessel; and third, the ability to reduce in weight and size when dried. These themes are discussed in more detail in Section 5. This signposted that the material’s characteristics directly informed the fermentation process, which could inform the production of filaments. This was explored in experiments which co-designed with bacterial cellulose characteristics described in the following section.

### 2.2. Materials and Methods

#### 2.2.1. Bacterial Cellulose Production Method

Fermentation of bacterial cellulose was prepared at the Centre for Novel Agricultural Products (CNAP) at the University of York. Bacterial cellulose is formed by bacteria secreting ribbons of cellulose. These ribbons form together at the surface of a nutrient-filled liquid and oxygen. Because of this, bacterial cellulose grows to the shape of the vessel it is contained in while maintaining its nonwoven-like structure. This method of forming BC is known as static fermentation.

Within this project, bacterial cellulose was produced from the strain Komagataeibacter xylinus ATCC 53524, obtained from the American Type Culture Collection (ATCC). The bacteria were grown as advised by ATCC. In brief, the medium used was the Hestrin-Schramm (HS) medium that contains 5 g/L bacterial peptone, 5 g/L yeast extract, 2.7 g/L Na_2_HPO_4_ and 1.5 g/L citric acid with the later addition of 20 g/L glucose. The glucose solution was sterile-filtered and added to the HS medium after it had been autoclaved to prevent the Maillard reaction, which occurs when autoclaving sugars and amino acids together. Cultures were statically incubated at 30 °C. Stocks of ATCC 53524 were prepared as previously described and stored at −70 °C with 25% glycerol [40]. This method of creating bacteria cellulose was continued across all experiments.

#### 2.2.2. Experimental Setup

The following sections describe the experiments and co-design method building on approaches of MDTD and co-design that led to the bioproduction of bacterial cellulose filaments.

#### 2.2.3. Conceptualisation of a Bacterial Cellulose Filament Growth Process

The initial conceptualisation of transforming bacterial cellulose growth into a filament began with research into established textile filament and yarn manufacturing processes. The textile manufacturing process of vortex spinning takes staple fibres and spins them into a yarn through a spiralled air flow in a vortex motion. First experiments explored how the micro fibril and ribbon growth of bacterial cellulose could be spun into filament emulating a similar vortex process. However, using a spinning method to form the bacterial medium in a liquid vortex lacked the control required for bacterial growth into a continuous filament shape, and the outcomes were irregular spherical shapes as the cellulose formed around itself, as it occurs in the shaken fermentation method of bacterial cellulose production. The physical characteristics and growth of bacterial cellulose had to be evaluated and understood further for design purposes. The second stage of experiments investigated approaches for co-designing with the bacterial cellulose characteristics. The material characteristics informed the next experiments.

#### 2.2.4. Development of the Co-Design Process

The second stage of experiments involved co-design with the material characteristics of bacterial cellulose. The initial experiments discussed above highlighted three physical designable characteristics: Due to the bacterial cellulose forming in a randomised web of interlocking cellulose ribbons, this set of experiments guided the growth into a filament shape by applying bacteria, separate from the nutrients to a specific pattern. First, the nutrient medium was solidified into a gel-like state by adding agar. Then, a filament shape was engraved into the agar, and the bacteria was added into the engraved shape. The growth across the agar medium was uneven, and often spores would bleed out of the contained engraving. The bacterial cellulose that formed adhered to the agar and thus could not be removed. This did, however, show that the bacteria would grow within a defined and small space with access to nutrients and oxygen.

#### 2.2.5. Method for Bioproduction of Bacterial Cellulose Filaments

By interacting with the material within a laboratory, the material characteristics were integrated in the co-design process. The bacterial cellulose requires a surface to form into a fibrous structure through access to nutrients and oxygen. However, the growth formation could be encouraged into a designed shape to change the physical parameters of growth. The following sections discuss the successful experiment in which the design was informed by the characteristics of the biological material.

#### 2.2.6. 3D-Printed Spiral

A reusable 3D-printed fermentation spiral was designed by Roberta Morrow and made by Mohamid H Hassen at the 3D printing and bioengineering Lab at the University of Manchester. The spiral was made from polylactic acid (PLA) with the intention of reuse in different experiments. With a 2 mm growth gap, 20 cm in width and 2 cm in height, this was designed to allow for shrinkage. Within the drying process, bacterial cellulose shrinks up to 80% in weight and size once the fluid has been removed. The aim was to initially create a yarn-like diameter with a length of 7 m (Figure 1). The vessel was sterilised using 70% ethanol and air dried in a laminar flow cabinet. The HS medium was seeded with ATCC 53,524 glycerol stocks (3% *v*/*v*) and then transferred to the spiral using a serological pipette. We prepared the inoculum as described by Florea et al., 2016, and measured the OD600 of the inoculum. We tested a range of loading and obtained optimised results using 3% (*v*/*v*) loading [40].

#### 2.2.7. Fermentation

Static fermentation was chosen to produce continuous filament growth to not disturb the entanglement of the microfilaments. A Petri dish lid was placed on top of the spiral to maintain sterile conditions before being transferred to a 30 °C room to incubate statically for 48 h. During the fermentation process, the bacterial cellulose formed at the air–liquid interface of the media, producing a cellulose pellicle which is filament-like. When wet, the filament has a diameter of 2 mm, and when dried this is reduced to 0.5 mm.

#### 2.2.8. Harvesting and Cleaning

The pellicle was lifted from the spiral using fine tweezers and carefully wrapped around a glass cylinder to prevent the cellulose string from intertwining. The glass cylinder was submerged in a glass beaker containing 0.1 M NaOH and heated at 80 °C for 5 h. This removed the bacteria from the pellicle. The cellulose was thoroughly washed in deionised water before air drying at room temperature. The spiral was thoroughly washed between each experiment. The filaments were lightly twisted to mimic a two-ply twisted yarn. This was then hand knitted into a small swatch using 2 mm knitting needles to create a small 3 cm by 5 cm swatch shown in the next section. The small, knitted textile test swatch demonstrated how the filament can be introduced into textile form.

## 3. Results

The aim of this project was to investigate whether filaments could be produced directly from the fermentation of bacterial cellulose using approaches of co-design with the material characteristics. The successful results of the filament prototype show that bacterial cellulose can directly produce filaments. By allowing the inherent characteristics of the material to have an influence on the design of the production process, an efficient production of filaments is demonstrated.

As a result of co-designing with the material, three designable characteristics were identified: firstly, the ability to self-assemble at the interface of oxygen and the nutrient medium; secondly, the link of physical growth and structural characteristics; and thirdly, the ability to reduce in weight and size depending on liquid content. It was found that these three functions point to the designability of the shape towards the bioproduction of filaments directly from fermentation.

The process of bacterial cellulose production was kept the same as pellicle growth to understand how the material interacts with physical changes. The filament production was successful in producing an even amount of cellulosic filament that could be replicated through the repeated use of the spiral. The filament can be seen dried in Figure 2a. Two ends of the filament were then twisted together, and a small fabric sample using hand knitting was produced, Figure 2b. The filament knitted well and demonstrated the stretch and recovery advantages of knitted form. As the production is directly from growth, additional manufacturing steps to form filament shape are reduced. Further, by growing into a filament, bacterial cellulose is opened to a wealth of new material characteristics such as knitted form.

## 4. Discussion

Bacterial cellulose is commonly produced in sheet form from static fermentation, which results in a biological nonwoven-like material [17]. Designers and scientists have explored ways of cutting out apparel manufacturing steps, such as pattern cutting, by utilising bacterial cellulose’s ability to form itself [16,17,33]. Although these examples are not defined as co-design specifically, the outcomes of these practices and manufacturing processes reflect the inherent characteristics of the bacterial cellulose material and can be used to inform production. A combination of material-driven textile design and human-centred co-design are considered to discuss the method of co-designing with biological materials.

A method of co-designing with material characteristics was applied to the production of bacterial cellulose filaments directly from fermentation. The outcomes of co-designing with bacterial cellulose are threefold: firstly, the research highlights three physical characteristics of bacterial cellulose that are inherently designable; secondly, this research shows that by co-designing with a biomaterial, efficient new production can be designed; thirdly, this paper has shown there is further space to investigate using co-design as a method for biomaterial production, bypassing conventional textile production steps and potentially shortening value chains.

### 4.1. What Are the Designable Characteristics of the Material That Aid Co-Design?

Although bacterial cellulose has a multitude of characteristics [24], this discussion highlights three designable characteristics that have been found within this research. The bacterial growth forms at the interface of liquid nutrients and oxygen. With the initial experiments, this was highlighted as a core function and vital to the successful growth of the biomaterial through fermentation. However, the two elements can be experimented with by designing where the nutrients and oxygen meet. This led to the second designable function, which is the physical characteristic of the pellicle. The physical properties of growth result in inherent mechanical characteristics such as strength. In order to maintain this characteristic of bacterial cellulose pellicles, the process of growth could not be separated from the production outcome. Thus, the growth process formation had to be a part of the design. The third designable characteristic of bacterial cellulose is its ability to reduce in size and weight as it dries. As the liquid evaporates from the material, the fibrils reduce in size and are less swollen. This is due to the multitude of hydroxyl bonds within the chemical structure of cellulose.

### 4.2. Co-Designing with the Characteristics of Bacterial Cellulose

Three physical elements that can be co-designed were stated above. The first characteristic was bacterial cellulose’s ability to grow within a confined environment but requiring access to nutrients and oxygen. The spiral restricts the formation of bacterial cellulose from a large surface area to a smaller surface area to change its physical form. This was achieved using a maze-like spiral structure. Secondly, it was highlighted that the process of the growth and the mechanical material characteristics such as strength cannot be separated. Thus, the ribbons of bacterial cellulose growth must be allowed to form in order to maintain the inherent qualities. The thickness of the growth determines the width of the filament; therefore, sufficient space is needed for the bacterial cellulose ribbons to entangle while controlling the thickness of the resulting filament. This leads to the third characteristic: the material changes in weight, texture and form when in a wet state, and this affects a reduction in size. This worked well combined with the above characteristic of requiring sufficient space to grow as it allowed the material to be produced with a larger diameter of 2 mm, which then reduced to 0.5 mm in diameter after drying. The inherent mechanical strength of bacterial cellulose was therefore maintained. These characteristics point to implementing a physical design of a spiral to navigate the production. This also highlights the relationship between the growth of the material and its characteristics. The outcome shows how co-designing with the material characteristics forms an understanding of the physical conditions that can be designed.

Further research could investigate the tensile strength of the filaments and characterisation towards their application in textile contexts. The filaments produced in this research were not fine enough to be tested on the filament tensile tester. This points to possibilities for further investigation of finer filaments in the bioproduction process. Finer filaments were not produced within this study as the paper focuses on demonstrating the production process and method of co-design. This can be expanded in future research. In addition to this, the filaments do not have the twist of a yarn to produce comparative results. Future research could produce filaments for evaluation against filaments and yarns for textile forms in a further investigation of the application of the filaments through testing.

## 5. Conclusions

This research investigated how co-designing with the characteristics of bacterial cellulose can inform new bioproduction processes and resulted in growing filaments directly from fermentation. The background section of this paper shows how innovative bioproduction sits at the intersection of biological sciences and material design. In order to utilise the biomaterial to the best of its ability, the material in question, namely bacterial cellulose, must be an integral part of the production decisions. This highlights how the material becomes an active part of the production method. A method for co-designing with the material’s characteristics was explored to form bacterial cellulose filaments directly from growth. This method was inspired by successful bioproduction outcomes found within the background review and built on material-driven textile design and human-centred co-design approaches.

Bacterial cellulose has successfully been translated into nonwoven materials by designing novel bio-manufacturing systems combining biology, design and textile manufacturing. These investigations have been designed around the fact that the bacterial cellulose grows like a biological nonwoven, consisting of micro filaments that entangle. This, however, leaves a gap to investigate, namely, introducing bacterial cellulose into a wealth of alternative textile elements and production processes such as textile filaments and yarns.

The characteristics of bacterial cellulose were explored through interdisciplinary collaboration, a literature review and physical experiments, which then directly informed a co-design method of filament production. The resulting filament prototype that was produced shows that the characteristics of bacterial cellulose can inform the bioproduction of textiles. This paper’s results highlight three designable characteristics: the ability to self-assemble, the link of growth with structural characteristics and the ability to reduce in weight and size after drying. By co-designing with these characteristics, filaments were produced directly from fermentation, making the need for further wet spinning processing for fibres and filament manufacturing steps redundant. The designable characteristics of bacterial cellulose discussed within this paper can be tuned further and developed to investigate the capabilities of the filaments. For example, textile testing to understand future applications, texturising of the filaments, finer filament growth and scaling of production are all future areas for investigation.

## Figures and Tables

**Figure 1 materials-16-04893-f001:**
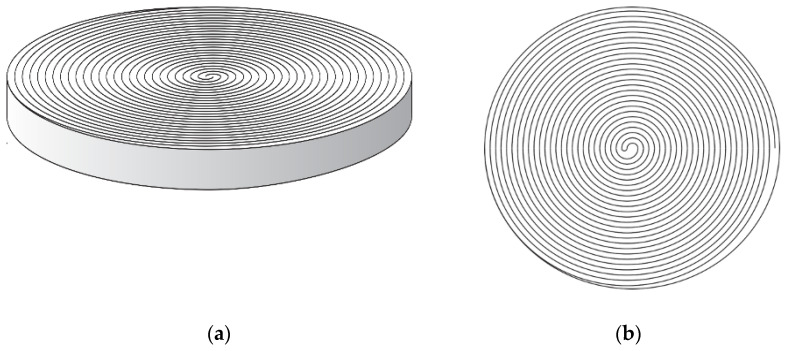
(**a**) Side view of spiral created; (**b**) bird’s eye view of spiral.

**Figure 2 materials-16-04893-f002:**
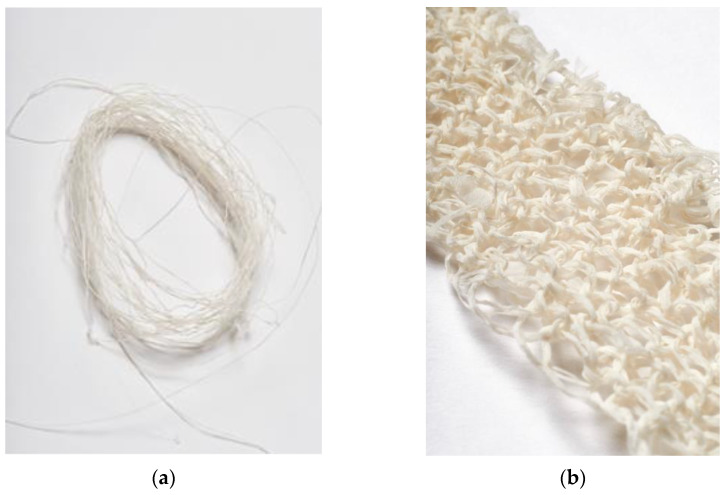
(**a**) 100% dried bacterial cellulose filament. (**b**) 100% dried bacterial cellulose filament in hand knitted structure.

## Data Availability

The data presented in this study are available on request from the corresponding author.

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
