# Peer review of "Bio-Producing Bacterial Cellulose Filaments through Co-Designing with Biological Characteristics"

_materials, 2023, doi:10.3390/ma16144893_

Round 1

Reviewer 1 Report

The experimental article "Bio-manufacturing bacterial cellulose filaments through co-designing with biological characteristics" is fully consistent with the theme of the publication "Polymers". The relevance of the work lies in research aimed at obtaining VS threads in a new way. In general, the work is relevant and will be of interest to a wide range of readers in connection with research conducted to reduce the cost of obtaining bacterial cellulose and using agricultural waste as nutrient media.

The strength of the work is the choice of the topical object of study - bacterial cellulose.Слабая сторона работа отражена ниже в замечаниях и вопросах рецензента:

1. The article says that the concentration of the inoculum introduced into the nutrient medium was (3% v/v). How was this concentration chosen, given that many studies use volume (10% v/v).

2. “Physically bacterial cellulose could be guided and designed to grow by restricting the flow of nutrients. This was done using a maze-like spiral structure." How can a spiral block nutrients when the nutrient medium is a liquid that fills the spiral labyrinth. Most likely, the spiral prevents the formation of a dense sheet of BC. Incorrect statement.

3. Why, when obtaining BC threads, a static cultivation method was used, and not a dynamic one. The dynamic culture method is used to obtain filaments and prevent the formation of a dense BC sheet.

4. Strength values for the resulting BC filaments after 48 hours of cultivation are not shown.

5. It is not clear for what purpose the use of nutrient media from agricultural waste is being discussed. It appears from the materials and methods that Hestrin-Schramm (HS) standard growth medium was used.

6. I recommend changing the title of the article "Bioproduction of bacterial cellulose filaments by co-design with biological characteristics".

Author Response

Response to Reviewer 1 Comments

Over all points - The experimental article "Bio-manufacturing bacterial cellulose filaments through co-designing with biological characteristics" is fully consistent with the theme of the publication "Polymers". The relevance of the work lies in research aimed at obtaining VS threads in a new way. In general, the work is relevant and will be of interest to a wide range of readers in connection with research conducted to reduce the cost of obtaining bacterial cellulose and using agricultural waste as nutrient media.The strength of the work is the choice of the topical object of study - bacterial cellulose.Слабаясторона работа отражена ниже в замечаниях и вопросах рецензента: 

Over all response

We thank the reviewer for the comments on the ‘Bio- Producing  bacterial cellulose filaments through co-designing with biological characteristics’ article. We have addressed the  comments below and edited the article accordingly.

Point 1: The following has been inserted into the paper under methods and materials: The HS medium was seeded with ATCC 53524 glycerol stocks (3% v/v) and then transferred to the spiral using a serological pipette. We prepared the inoculum as described by Florea et al, 2016 and measured the OD600 of the inoculum.  We tested a range of loading and obtained optimised results using 3% (v/v) loading.

Response 1: The article says that the concentration of the inoculum introduced into the nutrient medium was (3% v/v). How was this concentration chosen, given that many studies use volume (10% v/v).

Point 2: “Physically bacterial cellulose could be guided and designed to grow by restricting the flow of nutrients. This was done using a maze-like spiral structure." How can a spiral block nutrients when the nutrient medium is a liquid that fills the spiral labyrinth. Most likely, the spiral prevents the formation of a dense sheet of BC. Incorrect statement.

Response 2: The following has been changed into the paper under methods and materials:  The spiral restricts the formation of bacterial cellulose from a large surface area to a smaller surface area to change its physical form. This was done using a maze-like spiral structure.

Point 3: Why, when obtaining BC threads, a static cultivation method was used, and not a dynamic one. The dynamic culture method is used to obtain filaments and prevent the formation of a dense BC sheet.

Response 3: A series of experiments prier to obtaining the spiral production method has been added to add to the method section. This adds context as to how the final method was constructed. This includes  a type of shaken cultivation experiment, this experiment formed amorphous shapes and not filament shape.

Point 4: Strength values for the resulting BC filaments after 48 hours of cultivation are not shown.

Response 4: This has been edits and further a section as to why the material was not tested/ how it can be tested in the future has been added

Point 5: It is not clear for what purpose the use of nutrient media from agricultural waste is being discussed. It appears from the materials and methods that Hestrin-Schramm (HS) standard growth medium was used.

Response 5: Waste has been captured as  potential feedstock for bacterial cellulose, the following has been added to highlight that although this method discuses glucose as  the feedstock the process is directly translatable to  bacterial cellulose derived from waste. - Although this paper investigates  bacterial cellulose derived from glucose within the methodology, the methods are directly transferable to bio-waste derived bacterial cellulose, such as crop residues and the organic fraction of municipal waste, which is investigated with the co-authors of this article within the BBSRC Bio-manufacturing Textiles from Waste project.

Reviewer 2 Report

The manuscript really presents a nice idea for the production bacterial cellulose filaments. But author must focus on the following points:

1. What is the novelty of the work? 

2. The author have identified three questions to answer through their work as follows:.

a. What are the characteristics of the material?

b. How can the physical elements of the material growth be co-designed? and

c. what influence can textile manufacturing knowledge have within this? 

After reviewing the manuscript clear answers are not available. It is advisable that the questions may be answered when some variations will be made in the process of fabrication.

3. Does the filament have uniform crystallinity and strength throughout the total bundle. 

4. How it is useful in textile application or in any other. 

Author Response

Response to Reviewer 2 Comments

Over all response

Thank you for your review on the ‘Bio- Producing  bacterial cellulose filaments through co-designing with biological characteristics’ article (new working title).  It has been great to work through your points and very helpful to the development of the paper. Major  changes to this paper have been made centering the design methodology more to provided context  in adding experimental process into how the  final method was produced. Please see response to how we have implemented your comments below.

Point 1: What is the novelty of the work? 

Response 1: The novelty of this work is the exploration of growing bacterial cellulose directly into a fibre. This passes potential production routes such as wet spinning showing potential to reduce manufacturing steps. To achieve this a design method of co-designing with a living material was implemented. This has been added in detail throughout the paper.

Point 2: The author have identified three questions to answer through their work as follows:

  1. What are the characteristics of the material?
  2. How can the physical elements of the material growth be co-designed? and
  3. what influence can textile manufacturing knowledge have within this? 

After reviewing the manuscript clear answers are not available. It is advisable that the questions may be answered when some variations will be made in the process of fabrication.

Response 2: These questions have been removed from the paper structure. In response to this, the structure of the paper has changed becoming more concise in the method though to revised structure.  

Point 3: Does the filament have uniform crystallinity and strength throughout the total bundle. 

Response 3: The filament is not crystallin in structure as the natural characteristic of bacterial cellulose produces itself in a nonwoven structure. This has been added to the article in several sections discussing how BC is formed and how the filament has the same structure as the BC as a pellicle.  However, in future research, biological modifying of BC could investigate aligning the micro fibrils to make the material stronger.

Point 4: How it is useful in textile application or in any other. 

Response 4: this was not used in a textile application.  However, the paper looks to future research towards future application of textiles and suggests that by growing BC into fibres these avenues are opened. The sample was knitted into a swatch to conceptualise this.

Reviewer 3 Report

The author reported “Bio-manufacturing bacterial cellulose filaments through co-designing with biological characteristics”. Materials is a quality journal, the accepted manuscript must be significant, novel, innovative, and sufficient to scientific impact. Therefore, I suggest to improve this work for publication in this journal. My detailed comments and suggestions are follows:

First confusion is either it is a review article/experimental paper or book chapter, this work lack novelty to be considered as an original experimental paper.

Abstract: Re-write, provide some important methods and quantitative findings/results

In introduction, provide some more recent studies, discuss their research gaps

Section 2 of the manuscript is not needed in a research paper

Methods and materials: In experimental design, authors should use some statistical analysis, provide and compare with standard methods, provide related references.

Results: provide more data interms of tables and figures, For the morphologies, better to have light microscope pictures, SEM and TEM, add more studies special analytical instrumentation

State main findings only in conclusion, no need to provide references.

Author Response

Response to Reviewer 3 Comments

Overall response

Thank you for your review on the ‘Bio- Producing  bacterial cellulose filaments through co-designing with biological characteristics’ article (new working title).  It has been great to work through your points and very helpful to the development of the paper. Major  changes to this paper have been made centering the design methodology more to provided context  in adding experimental process into how the  final method was produced. Please see response to how we have implemented your comments below.

Point 1: First confusion is either it is a review article/experimental paper or book chapter, this work lack novelty to be considered as an original experimental paper.

Response 1: The novelty of this work is the exploration of growing bacterial cellulose directly into a fibre. This passes potential production routes such as wet spinning showing potential to reduce manufacturing steps. To achieve this a design method of co-designing with a living material was implemented. This has been added in detail throughout the paper. The paper has been majorly edited to centre the method in with the experiments.

Point 2: Abstract: Re-write, provide some important methods and quantitative findings/results

Response 2: Abstract has been re-written providing context on how the method was produced and outcomes.

Point 3: In introduction, provide some more recent studies, discuss their research gaps.

Response 3: Added context into design studies and theories within the introduction. The questions previously contextualising the paper have been taken out and made more coherent in how design was implemented in this process though co-creation.

Point 4: Section 2 of the manuscript is not needed in research paper.

Response 4: background has been incorporated into the introduction as it contextualises the research.

Point 5: Methods and materials: In experimental design, authors should use some statistical analysis, provide and compare with standard methods, provide related references.

Response 5: methods section has been expanded majorly to include how the final method was produced. There is an active combination of design and science methods   within this paper showing the hybrid of knowledge used, this has been expanded. In terms of comparing against textile standards, this has been addressed in future research at the end of the dissection. ‘Further research could investigate the tensile strength of the filaments and characterisation towards their application in textile contexts. The filaments produced in this research were not fine enough to be tested on the filament tensile tester. This points to possibilities for further investigation of finer filaments in the bio-production process. Finer filaments were not produced within this study as the paper demonstrates the production process and method of co-design. This Can be expanded in future research. In addition to this, the filaments do not have the twist of a yarn to produce comparative results.  Future research could produce filaments for evaluation against filaments and yarns for textile forms.’

Point 6: Results: provide more data interms of tables and figures, For the morphologies, better to have light microscope pictures, SEM and TEM, add more studies special analytical instrumentation

Response 6: There is an active combination of design and science methods   within this paper showing the hybrid of knowledge used, this has been expanded, this paper has been recentred and highlights the method of co-design to bio-produce filaments directly from fermentation. Further testing is encouraged for future research as above.

Point 7: State main findings only in conclusion, no need to provide references.

Response 7: references have been removed from conclusion.

Reviewer 4 Report

This is an interesting work which discuss a novel idea to produce BC filament.  Overall manuscript is well written and relevant literature is thoroughly discussed . However in my opinion the current manuscript resembles more like a concept paper instead of a research work. Experimental part is very short and many questions arise about the feasibility and reproducibility of the concept presented. I would suggest to add minor details about the apparatus designed and how the natural/conventional growth of BC pellicles at air/liquid interface could be avoided. 

Samples produced weren't characterised for morphological or related properties, which leads towards argument based discussion, supported with insufficient technical investigations/results.

No doubt the concept presented here is novel but the manuscript in current form could be published as a concept paper instead. 

Author Response

Response to Reviewer 4 Comments

Overall response

Thank you for your review on the ‘Bio- Producing bacterial cellulose filaments through co-designing with biological characteristics’ article (new working title).  It has been great to work through your points and very helpful to the development of the paper. Major changes to this paper have been made centering the design methodology more to provided context in adding experimental process into how the  final method was produced. Please see response to how we have implemented your comments below.

Point 1: I would suggest adding minor details about the apparatus designed and how the natural/conventional growth of BC pellicles at air/liquid interface could be avoided. 

Response 1:  More detail into the method and design of the spiral. The following has also been edited into the paper in the methods section and materials: The spiral restricts the formation of bacterial cellulose from a large surface area to a smaller surface area to change its physical form. This was done using a maze-like spiral structure.

Point 2: Samples produced weren't characterised for morphological or related properties, which leads towards argument-based discussion, supported with insufficient technical investigations/results.

Response 2: this has now been addressed within the discussion section, further stating how future testing can be a part of future research.  ‘Further research could investigate the tensile strength of the filaments and characterisation towards their application in textile contexts. The filaments produced in this research were not fine enough to be tested on the filament tensile tester. This points to possibilities for further investigation of finer filaments in the bio-production process. Finer filaments were not produced within this study as the paper demonstrates the production process and method of co-design. This Can be expanded in future research. In addition to this, the filaments do not have the twist of a yarn to produce comparative results.  Future research could produce filaments for evaluation against filaments and yarns for textile forms.’

Point 3: No doubt the concept presented here is novel but the manuscript in current form could be published as a concept paper instead. 

Response 3: we apricate the acknowledgement of the novelty of the work, the structure of the paper has changed majorly implementing more context on the method especially.

Round 2

Reviewer 1 Report

Accept the article for publication

Reviewer 2 Report

Now the aim of the work is clear.

Reviewer 3 Report

The authors have made sufficient modifications, and I suggest that this paper be accepted for the publication.